# Clinical Effectiveness of Fluorescence Lymph Node Mapping Using ICG for Laparoscopic Right Hemicolectomy: A Prospective Case–Control Study

**DOI:** 10.3390/cancers15204927

**Published:** 2023-10-10

**Authors:** Gyung Mo Son, Mi Sook Yun, In Young Lee, Sun Bin Im, Kyung Hee Kim, Su Bum Park, Tae Un Kim, Dong-Hoon Shin, Armaan M. Nazir, Gi Won Ha

**Affiliations:** 1Department of Surgery, Pusan National University Yangsan Hospital, Pusan National University School of Medicine, Yangsan 50612, Republic of Korea; kh8462@hanmail.net; 2Research Institute for Convergence of Biomedical Science and Technology, Pusan National University Yangsan Hospital, Yangsan 50612, Republic of Korea; msyun@pusan.ac.kr (M.S.Y.); vmffkdl38@naver.com (I.Y.L.); 3Department of Medicine, Pusan National University School of Medicine, Yangsan 50612, Republic of Korea; brandonim@naver.com; 4Department of Internal Medicine, Pusan National University Yangsan Hospital, Pusan National University School of Medicine, Yangsan 50612, Republic of Korea; psubumi@hanmail.net; 5Department of Radiology, Pusan National University Yangsan Hospital, Pusan National University School of Medicine, Yangsan 50612, Republic of Korea; kimtaeun78@hanmail.net; 6Department of Pathology, Pusan National University Yangsan Hospital, Pusan National University School of Medicine, Yangsan 50612, Republic of Korea; donghshin@chol.com; 7School of Medicine, Royal College of Surgeons in Ireland, University of Medicine and Health Sciences, D02 YN77 Dublin, Ireland; armaannazir15@gmail.com; 8Research Institute of Clinical Medicine of Jeonbuk National University-Biomedical Research Institute, Jeonbuk National University Hospital, Jeonju 54907, Republic of Korea; acts29@jbnu.ac.kr

**Keywords:** colonic neoplasms, hemicolectomy, fluorescence, indocyanine green, lymph nodes, lymphatic metastasis

## Abstract

**Simple Summary:**

Distinguishing D3 lymph nodes and actual lymphatic pathways from a primary tumor can be difficult during surgery, making it challenging to confirm the completeness of D3 lymph node dissection. Fluorescence lymph node mapping (FLNM) is a promising method for lymph node visualization. FLNM significantly increased the harvested D3 lymph node count by 50% and enhanced the detection rate of metastatic D3 lymph nodes by two times in stage III colon cancer. Consequently, FLNM can be effective in improving the oncological quality and completeness of D3 lymph node dissection in patients with stage III colon cancer during laparoscopic right hemicolectomy.

**Abstract:**

Background: The distinction between D3 lymph nodes and actual lymphatic pathways in primary tumors can be difficult during surgery, making it challenging to confirm the completeness of D3 lymph node dissection. Fluorescence lymph node mapping (FLNM) is a promising method for lymph node visualization. Purpose: This study aimed to assess whether FLNM enhances the effectiveness of D3 lymph node dissection in patients with right-sided colon cancer. Methods: Endoscopic submucosal indocyanine green injection were performed on the distal margin of the colon cancer. In an FLNM group, the lymphatic drainage pathway and distribution of D3 lymph nodes were explored. Pathological evaluations were conducted for the fluorescent D3 and non-fluorescent D3 lymph nodes. Results: The FLNM group showed a significantly higher number of harvested lymph nodes in the D3 area. In stage III patients, the proportion of D3 lymph node metastasis was significantly higher in the FLNM group. The harvested D3 lymph node count showed a proportional correlation with a metastatic lymph node count of up to 15. Conclusion: FLNM could be considered a promising new strategy to potentially increase harvested D3 lymph node counts in colon cancer surgery.

## 1. Introduction

Complete resection of a primary tumor and its related lymph nodes is crucial for the surgical treatment of colon cancer. The number of retrieved lymph nodes, which has a significant impact on long-term oncological outcomes in advanced colorectal cancer, is important in achieving radical lymph node dissection. Lymph node dissection extent is of practical interest to colorectal surgeons [1,2,3]. The Japanese Society for Cancer of the Colon and Rectum (JSCCR) guidelines recommend D3 lymph node dissection for stage II and III colorectal cancers, and D2 dissection for early-stage disease [4]. Complete mesocolic excision (CME) with central vessel ligation is conceptually similar to the D3 lymph node dissection in Japan [5]. Therefore, the removal of D3 lymph nodes at the origin of feeding vessels connected to the primary tumor is widely accepted as an essential component of oncological surgery for colon cancer. Consequently, D3 lymph node dissection is recommended as a part of radical resection for locally advanced colon cancer.

However, adequate D3 lymph node dissection in patients with right-sided colon cancer is technically challenging. The distinction between D3 lymph nodes and the actual lymphatic pathways from the primary tumor can be difficult during surgery, making it challenging to confirm the completeness of D3 lymph node dissection [6]. In highly immunogenic colorectal cancers, such as those with unstable microsatellite instability, an active anti-tumor immune response can lead to pronounced peritumoral lymphocyte infiltration. Performing surgery on such immunogenic tumors may result in an increased harvest of non-metastatic lymph nodes. While the therapeutic benefits of D3 lymph node dissection for early-stage colon cancer patients may be limited, the increased harvest of non-metastatic lymph nodes could serve as a potential indicator of a relatively favorable prognosis for these immunogenic tumors [7].

Complete removal of all metastatic D3 lymph nodes can provide an opportunity for adjuvant therapies to improve survival in patients with stage III colon cancer [8]. Therefore, an intraoperative visual assessment that allows easy identification of D3 lymph nodes and lymphatic drainage pathways would be of great value in enabling surgeons to perform optimal oncological surgeries. One method of lymph node visualization is fluorescence lymph node mapping (FLNM) using indocyanine green (ICG). With the recent advancements in near-infrared (NIR) camera systems for minimally invasive surgery, the use of fluorescence imaging in colorectal surgery has increased [9,10,11]. Following injection into peritumoral areas, ICG spreads through the lymphatic flow and binds to macrophages within the lymph nodes. Fluorescence lymph node mapping allows real-time visualization of lymph nodes and the lymphatic drainage pathways during surgery. However, some studies have evaluated the feasibility of FLNM for right-sided colon cancer, and there is insufficient evidence regarding its role in achieving adequate lymph node dissection [12,13,14,15].

This study aimed to assess whether FLNM enhances the effectiveness of D3 lymph node dissection in patients with right-sided colon cancer.

## 2. Materials and Methods

### 2.1. Patients

We enrolled 291 patients who underwent laparoscopic right hemicolectomy between January 2018 and August 2022 at the Pusan National University Yangsan Hospital, Korea. Inclusion criteria were patients with right-sided colon cancer of adenocarcinoma with invasion depth beyond the submucosal layer between 19–80 years of age without distant metastasis or other organ cancer. Right-sided colon cancers were defined as tumors located in the cecum, ascending colon, hepatic flexure colon, or proximal transverse colon. Exclusion criteria were patients <19 or >80 years of age (*n* = 6), patients diagnosed with a pathology other than adenocarcinoma, such as adenoma or adenocarcinoma in situ (*n* = 54), those who underwent combined resection of other organs (*n* = 11) or emergency surgery (*n* = 10), and those who had distant metastasis at the time of first diagnosis (*n* = 7). Of the 203 patients included in the study, 73 underwent FLNM, and 130 underwent conventional radical surgery without FLNM (Figure 1). Institutional Review Board approval (IRB No. 05-2018-182) was obtained for this study. Written informed consent was obtained from all the included patients.

### 2.2. Fluorescence Lymph Nodes Mapping 

Endoscopic tattooing was performed using ICG (DIAGNOGREEN INJ. 25 mg, manufactured by Daiichi Sankyo, Tokyo, Japan) for the FLNM. The concentration and total amount of ICG solution were 0.25 mg/mL and 1 mL, respectively. This was injected into the submucosal layer during the endoscopic procedure. The concentration and dosage were determined based on a previous study aimed at optimizing a fluorescence protocol for successful ICG angiography, fluorescent tumor localization, and FLNM [8,16]. The endoscopic submucosal ICG injection procedure, tattooing, was performed at two sites, as close as possible to the distal margin of the colon cancer; 0.5 mL of ICG solution was applied to each tattooing site. Multiple tattooing sites were chosen to enhance the accuracy and reliability of the FLNM because colon cancer can have diverse lymphatic drainage pathways, with lymphatic channels potentially leading to different regional and apical lymph nodes. The FLNM can be expected to cover a broader range of potential lymphatic drainage pathways by selecting multiple tattooing sites near the distal margins of colon cancer. The endoscopic submucosal ICG injection procedure involved injecting ICG within the submucosal layer of the colon 12–18 h before the scheduled surgery. The timing of the endoscopic tattooing procedure performed a day before surgery allowed sufficient time for the ICG to drain through the submucosal lymphatic channels. By remaining in the submucosal layer, ICG facilitated visualization of the tumor location using a laparoscopic NIR camera (1588 AIM camera system, Stryker, Kalamazoo, MI, USA). ICG drains to the regional lymph nodes (D1) and passes through intermediate lymph nodes (D2) in the colonic mesentery, finally reaching the apical or D3 lymph nodes as gateways to the systemic lymphatic pathway. The distribution of the ICG-stained regional and apical lymph nodes was explored under their visualization as fluorescent lymph nodes using a laparoscopic NIR camera to determine the D3 lymph node dissection area. To assess the fluorescent lymphatic drainage pattern, the fluorescent marking of the primary tumor was first identified. Subsequently, the direction of fluorescent lymph nodes and the connecting lymphatic vessels was explored, and this was compared with the branching positions of the colonic artery. The connection of pericolic lymph nodes through lymphatic vessels to the ileocolic artery (ICA) lymph node (203) was designated as the ICA route, while their connection to the middle colic artery (MCA) lymph node (223) was designated as the MCA route. If the lymphatic drainage occurred to both the ICA and MCA lymph nodes, it was assessed as a dual route. This process allowed for the evaluation of the fluorescent lymphatic drainage pathway (Figure 2).

### 2.3. FLNM-Guided D3 Lymph Node Dissection

All the patients underwent laparoscopic right hemicolectomy with D3 lymph node dissection. The JSCCR classification was used for lymph nodes. Further, D3 lymph node dissection was defined as the removal of the pericolic lymph nodes (D1), intermediate lymph nodes (D2), and main or apical lymph nodes (D3). Lymph node dissection was performed on the anterior surface of the superior mesenteric vein (SMV). Mesenteric dissection was performed cephalically to explore the root of ICA, right colic artery (RCA), and MCA. Sharp dissection was performed to remove lymphatic tissues at the root of the arteries that covered the anterior surface of the SMV. In an FLNM group, the fluorescent lymphatic drainage pathway and distribution of D3 lymph nodes were explored prior to the lymph node dissection. The medial extent of D3 lymph node dissection was determined based on the identification of fluorescent lymph nodes, which was set along a dissection line that allowed for the removal of all fluorescent lymph nodes identified in the roots of the ICA, RCA, and MCA. If fluorescent lymph nodes were found behind the MCA root of the superior mesenteric artery (SMA), deep lymphoadipose tissues between the posterior area of the MCA and the inferior margin of the pancreas were removed. After a radical D3 lymph node dissection, the absence of any residual fluorescent lymph nodes was confirmed to ensure that all identified fluorescent lymph nodes were successfully removed. This step was performed to assess the completeness of lymph node dissection and to reduce the chances of leaving behind any potentially involved lymph nodes. In the control group, the medial extent of the D3 lymph node dissection was determined based on anatomical landmarks. The medial boundary was set along an imaginary line based on a gap between the SMV and SMA. The MCA root was identified and dissected into its right branch.

In the CME procedure, complete mobilization of the mesocolon was performed by dissecting the Toldt’s fascia from the retroperitoneum to achieve complete removal of an intact package, consisting of the tumor and its associated lymphatic drainage within the mesentery. In both groups, the terminal ileum was divided approximately 5–10 cm away from the ileocecal valve. The distal longitudinal resection margin extended at least 5–10 cm from the tumor. For cancers of the cecum and ascending colon, the right branch of the MCA was divided. In cases of cancer involving the hepatic flexure and the right third of the transverse colon, central ligation of the MCA was performed.

### 2.4. Pathologic Evaluation

After completing the surgical procedure, including the radical D3 lymph node dissection, the specimen was extracted using a transumbilical mini-laparotomy incision. In the FLNM group, the surgeon carefully harvested the fluorescent D3 lymph nodes under guidance and visualization provided by the laparoscopic NIR camera. The lymph nodes were labeled them fluorescence D3 lymph nodes, according to their anatomical association with the colic arteries, including ICG-ICA (203), RCA (213), or MCA (223) lymph nodes. When non-fluorescent D3 lymph nodes were identified in the remaining lymphoadipose tissue near the vascular root, they were separated and labeled as non-fluorescent ICA, RCA, or MCA lymph nodes. In the control group, excised colonic tissue was placed on an auxiliary table, and the lymph nodes around the ligation sites of the colonic arteries were visually examined with the naked eye. The D3 lymph nodes around the vascular ligation sites were dissected and separated from the surrounding tissues without additional fluorescence imaging techniques. The D3 lymph nodes were labeled as ICA, RCA, or MCA lymph nodes.

A pathologist evaluated the lymph nodes obtained from each labeled tissue specimen in a laboratory. They specifically isolated the pericolic and intermediate lymph nodes buried within the mesocolon by manual palpation. The harvested lymph nodes were divided into two categories: pericolic/intermediate and D3 area lymph nodes. Pathological evaluations were conducted separately for the labeled fluorescent D3 and non-fluorescent D3 lymph nodes. Hematoxylin and eosin staining was performed to assess the presence of metastatic lymph nodes. The pathologist counted the number of harvested and metastatic lymph nodes in the pericolic/intermediate and D3 lymph node regions.

### 2.5. Clinical Outcomes

To determine the effectiveness of FLNM in D3 lymph node dissection during laparoscopic right hemicolectomy, the number of harvested and metastatic lymph nodes was compared between the FLNM and control groups. The number of D3 lymph nodes, particularly in the ICA and MCA roots, was evaluated. Since the RCA was not commonly found in most patients, those with both ICA and RCA lymph nodes were combined and classified as ICA lymph nodes for analysis. In the FLNM group, the lymphatic drainage pathway was assessed by dividing it into the ICA and MCA routes, and the location of the metastatic lymph nodes was compared with that of the lymphatic drainage pathway. Additionally, we attempted to determine the factors related to the number of harvested and metastatic lymph nodes in patients who underwent laparoscopic right-sided hemicolectomy with D3 lymph node dissection.

### 2.6. Statistical Analysis

Statistical analyses were performed to compare outcomes between the FLNM and conventional groups. The primary outcomes were the count of harvested/metastatic lymph nodes and the detection rate of metastatic D3 lymph nodes. The Student’s t-test was used to compare the mean values of harvested and metastatic lymph node counts. For the analysis of categorical variables related to the detection rate of metastatic D3 lymph nodes, Pearson’s Chi-square test and Fisher’s exact test were employed. To evaluate clinicopathologic factors that might influence the continuous variable of the count of harvested and metastatic lymph nodes, the correlation analysis was used, and the correlation plot was produced based on the Spearman correlation coefficient between the lymph node counts and corresponding clinical factors. The simple linear regression was conducted as a univariate analysis to identify relevant clinicopathologic variables to the lymph node counts. The multivariable linear regression was performed to assess the clinicopathologic factors relating to the lymph node counts as a multivariate analysis. Statistical analysis was performed using SPSS 27.0 (Statistical Package for Social Science Version 27.0, IBM SPSS, Armonk, NY, USA) and R software (version 4.3.0, R Foundation for Statistical Computing, Vienna, Austria), and the significance level was two-tailed at *p* < 0.05.

## 3. Results

### 3.1. Patients

The clinicopathological characteristics of the patients were comparable between the FLNM and control groups (Table 1). No adverse events were associated with the endoscopic submucosal ICG injection. The optimal FLNM protocol had a success rate of 98.6%. One male patient who experienced FLNM failure had high body mass index (BMI), thick mesenteric fat tissue, and locally advanced hepatic flexure colon cancer, resulting in compromised fluorescent lymphatic drainage.

### 3.2. Fluorescent Lymphatic Drainage Pathway

Schematic diagrams were used to illustrate the distribution of lymph node pathways and lymph node metastasis, according to the location of colon cancer in the FLNM group (Figure 3). The lymphatic drainage pathway patterns showed significant differences, depending on the location of the colon cancer (*p* < 0.001) (Table 2). A dual lymphatic drainage pathway, including the ICA and MCA routes, was observed in 22.2–44.2% of cases, depending on the location of the colon cancer.

### 3.3. Fluorescence Lymph Node Mapping and Metastatic Lymph Nodes

Among factors evaluating surgical quality, the FLNM group had a significantly higher number of harvested lymph nodes in the D3 area. However, there was no statistically significant difference in the metastatic lymph node counts when compared to the control group (Figure 4).

In the patients with stage III disease, the proportion of D3 lymph node metastasis was significantly higher in the FLNM group than in the control group (62.5% vs. 26.8%, *p* = 0.012). In particular, the proportion of patients with ICA lymph node metastasis was more than three times higher in the FLNM group than in the control group (Table 3).

### 3.4. Correlation of Harvested and Metastatic Lymph Nodes

In the FLNM group, there was a significant correlation between the lymphatic drainage pattern and distribution of D3 lymph node metastasis (*p* = 0.041) (Table 4). Even in cases without pericolic lymph node metastasis, D3 lymph node metastasis was observed in two (3.4%) patients. When pericolic lymph node metastasis was present, there was a significant correlation between the lymphatic drainage pattern and distribution of D3 lymph node metastasis (*p* = 0.039).

In the patients with metastatic D3 lymph nodes (*n* = 10), metastases were exclusively detected in the fluorescent D3 lymph nodes in 50% of cases (*n* = 5) and exclusively in non-fluorescent D3 lymph nodes in 20% of cases (*n* = 2). The proportion of metastatic fluorescent and nonfluorescent D3 lymph nodes was 30% (*n* = 3) (Table 5).

Regression analysis revealed a statistically significant correlation between the harvested and metastatic lymph node counts in patients with stage III disease (*p* < 0.001). The harvested D3 lymph node count showed a proportional correlation with a metastatic lymph node count of up to 15. However, beyond the 15 lymph nodes, the proportional correlation was reversed (Figure 5).

### 3.5. Regression Analysis for FLNM

Lymph node count was associated with various clinicopathological variables. The harvested lymph node count was influenced by the tumor size, resected specimen length, tumor aggressiveness, and operation time. The harvested D3 lymph node count was influenced by FLNM, young age, female sex, low BMI, and operation time. However, the metastatic D3 lymph node count was associated with tumor aggressiveness (Figure 6).

A simple and multivariable linear regression model was used to analyze the clinicopathological factors associated with lymph node count (Table 6). The FLNM had an impact that increased the harvested lymph node count by five times (*p* < 0.001). In the multivariable linear regression analysis, FLNM still had an independent impact that could increase the harvested lymph node count by four times (*p* < 0.001). However, the FLNM did not have a significant impact on the increase in metastatic lymph node count (*p* = 0.152). In stage III patients, lymphatic invasion was analyzed as an independent clinicopathologic factor related to the metastatic D3 lymph nodes (Table 7).

## 4. Discussion

Radical D3 lymph node dissection is recommended as a standardized surgical procedure for improving survival outcomes in patients with advanced-stage right-sided colon cancers [17]. Appropriate removal of D3 lymph nodes is essential for staging disease progression and establishing the most suitable treatment strategy for locally advanced colon cancer [18,19,20]. Therefore, the identification of the region of potentially involved D3 lymph nodes is crucial for effective oncological surgery. Previous studies have demonstrated that FLNM could be beneficial for adequate D3 lymph node dissection in colorectal cancer [8,9,10,11,12,13,14,15,21]. Therefore, FLNM may play a role in optimal D3 lymph node dissection during laparoscopic surgery for locally advanced colon cancers.

In this study, there were no statistical differences in terms of surgical resection specimen length, CME quality, and the mean levels of D3 lymph node metastases between the FLNM and control groups (Appendix A). This can be confusing when evaluating the usefulness of FLNM; it is disappointing that FLNM did not result in the discovery of more metastatic lymph nodes. One possible interpretation is that, even without FLNM, adequate D3 dissection was performed, leading to sufficient detection of metastasis in the D3 lymph nodes. In a landmark study comparing colon cancer patients with a harvested lymph node count of 12 versus those with a lymph node count <6, a significant difference in disease-free survival was observed [22]. The likelihood of detecting metastatic lymph nodes increased as the harvested lymph node count reached 12; beyond that threshold, there was no further increase in the odds ratio for node positivity. Therefore, the adequacy of radical lymph node dissection is currently evaluated based on a harvested lymph node count of 12. In this study, all the patients obtained ≥12 lymph nodes, meeting the usual criteria for lymph node radicality. Another possible explanation is that the percentage of metastatic lymph nodes among the harvested D3 lymph nodes was very low (only 2.5%), indicating that even with the use of FLNM to harvest more D3 lymph nodes, the number of metastatic lymph nodes may not show a statistical difference. Therefore, if we are already performing adequate D3 lymph node dissection through conventional surgery and there is no statistical difference in the number of metastatic lymph nodes, there may be no need to insist on conducting additional D3 lymph node dissection using FLNM.

Interestingly, the harvested lymph node count was directly proportional to the metastatic lymph node count in stage III patients. We were curious about the underlying reasons for the increase in the detection rate of D3 lymph node metastases using FLNM in stage III patients. The harvested D3 lymph node count of the FLNM group was on average 1.5 times higher than that of the control group. However, when removing more than 15 lymph nodes of the D3 area, the correlation of the count of metastatic lymph nodes became reversed. Therefore, in stage III patients, utilizing FLNM to dissect up to 15 lymph nodes of the D3 area is expected to enable optimal D3 lymph node dissection, ensuring robust detection of metastatic D3 lymph nodes without surgical compromise.

During this research, the D3 lymph node dissection was performed for all the patients. However, the harvested D3 lymph node counts were significantly higher in the FLNM group. There were some differences in the method used to determine the inner extent of D3 lymph node dissection between the FLNM and control groups. In the control group, the medial extent of D3 lymph node dissection was determined based on anatomical landmarks of an imaginary line between the SMV and SMA. The deep lymphoadipose tissue between the MCA root and the pancreas was not routinely removed. In the FLNM group, the medial extent of D3 lymph node dissection was determined based on the identification of fluorescent lymph nodes and a mesenteric dissection line that allowed for the removal of all fluorescent lymph nodes identified in the roots of the ICA and MCA. If fluorescent ICA lymph nodes were found over SMV and SMA, mesenteric dissections were extended over an SMV line. Fluorescent lymph nodes behind the MCA root were also dissected to the inferior margin of the pancreas. These differences in the extent of D3 lymph node dissection could affect the higher number of harvested D3 lymph node counts in the FLNM group. There is no evidence to support the removal of lymph nodes beyond the scope of conventional D3 dissection utilizing anatomical landmarks for nonmetastatic colon cancer. However, there is still a lack of definite consensus regarding anatomical landmarks and clearance level of D3 lymph node dissection. So, this interesting finding of a higher detection rate of metastatic ICA nodes in the FLNM group suggests the need for further research.

Another reason for the different lymph node counts could be related to the pathology laboratory. Although this study adhered to standardized pathologic evaluation procedures, the labeled fluorescent lymph nodes were delivered separately to the pathology laboratory. Consequently, the pathologists may have meticulously explored and identified more lymph nodes. This could explain the increased node count observed in the FLNM group.

The number of harvested D3 lymph nodes may depend on various factors, including the quality of surgical resection, tumor aggressiveness, and patient factors [23]. An interesting finding of this study was the different factors influencing the harvested and metastatic D3 lymph node counts. Younger age, female sex, and lower BMI were associated with successful D3 lymph node dissection and acquisition of a greater number of D3 lymph nodes. Additionally, FLNM was found to be a meaningful approach for obtaining D3 lymph nodes more effectively. However, metastatic lymph node count was more closely associated with tumor aggressiveness than other patient factors.

The fact that FLNM may not detect all metastatic lymph nodes highlights the need for caution when using it as the sole basis for determining the extent of lymph node dissection [24]. Lymph nodes filled with tumor cells cannot be detected by FLNM, so there are limitations in its ability to identify metastatic lymph nodes. The penetration of ICG into lymph nodes that are occupied by tumor cells can be blocked, limiting the formation of fluorescent images. Fortunately, it is often observed that non-fluorescent metastatic lymph nodes, which are not stained by FLNM, tend to have large size and irregular surfaces, making them more accessible for exploration through the visual and tactile senses of experienced surgeons. 

Additionally, the possibility that 20% of metastatic lymph nodes remain undetected using FLNM raises doubts regarding the limitations of its utility [25]. Among the D3 lymph node metastasis, 80% had metastasis confirmed in fluorescent lymph nodes, whereas only 20% had metastasis confirmed in non-fluorescent lymph nodes (Appendix A). Non-fluorescent metastatic lymph nodes could not be stained by the ICG solution because of the presence of cancer cells filling the lymph nodes, thereby preventing lymphatic flow. In such cases, the nodes were often >1 cm, firm in texture, had irregular surfaces, and occasionally adhered to the surrounding mesenteric tissue. Experienced surgeons can easily identify them through visual and tactile inspection. The metastatic fluorescent lymph nodes showed cancer cell infiltration within the lymph nodes, but the unoccupied lymphatic tissue allowed lymphatic flow. Therefore, combining FLNM with the visual inspection of experienced surgeons could be a better approach to radical D3 lymph node dissection. In particular, the completeness of the D3 lymph node dissection could be objectively evaluated using fluorescent lymph nodes. Therefore, FLNM can be beneficial for safely and efficiently performing D3 lymph node dissection, providing surgeons with objective indicators for assessing the completeness of D3 lymph node removal. 

This study has several limitations. First, the endoscopic submucosal ICG injection procedure for FLNM was performed one day before surgery. So, the FLNM had limitations in identifying sentinel nodes. If the aim is to explore sentinel nodes or to observe dynamic lymphatic drainage, it would be more advantageous to administer ICG submucosal or subserosal injections during the surgery [26,27,28]. Second, there were instances in which distinguishing between the D3 and D2 regions was ambiguous. The definition of D3 lymph nodes was limited to the areas around the branching points of major blood vessels; however, in cases where lymph nodes were continuously connected side by side, it was challenging to differentiate between D2 and D3 lymph nodes. In this study, D2 lymph nodes were not analyzed separately, and lymph nodes close to the tumor were counted together with pericolic lymph nodes, whereas lymph nodes in proximity to major vascular roots were counted as D3 lymph nodes. Third, the endoscopic ICG injection was applied at two locations along the distal margin of the tumor, which may not reflect lymphatic drainage from the proximal part of the tumor. This could lead to inadequate D3 lymph node staining in locally advanced colon cancers. Fourth, ICG is a cancer non-specific fluorescent dye, which limits its utility as a means of differentiating metastatic lymph nodes. By utilizing intravenous injections of cancer-specific fluorescent dyes instead of submucosal ICG injections, lymph node staining through the arterial flow is expected to be possible, even in metastatic lymph nodes with blocked lymphatic flow [29,30]. Fifth, there was a clear difference in the timing of enrollment between the two patient groups. During the initial phase, FLNM was performed only in a subset of patients to establish FLNM protocol standardization, while the majority underwent standard surgery. However, after establishing the standardized FLNM protocol, FLNM was implemented in all eligible patients. During the early period of the study (2018–2019), the FLNM group accounted for 15.5% of the cases, whereas in the late period (2020–2022), it increased to 90.9%. The difference in the timing of surgery between these two groups could potentially be influenced by variations in the surgical experience of the surgeon. Fortunately, the surgeon (SGM) had already accumulated over 10 years of experience in colorectal surgery when the study commenced, thus having overcome the learning curve. As a result, there was no difference in the quality of surgery between the FLNM and control groups (Appendix A). Finally, the number of stage III patients is insufficient for robust statistical confirmation in this study. The hypothetical efficacy of FLNM in stage III patients was estimated from a small sample size of a single institution. Based on these results, the statistical sample size was calculated as 180 patients for a randomized clinical trial. As a follow-up study, we designed a multi-institutional research project to evaluate the clinical efficacy of FLNM for the detection of D3 lymph nodes. Based on the results of this study, the total sample size was calculated as 180 for stage III patients.

## 5. Conclusions

FLNM could be considered a promising new strategy to potentially increase harvested D3 lymph node counts in colon cancer surgery. The use of FLNM has the potential to change the paradigm of oncological surgery, but the clinical benefits and oncological advantages are not yet clear. Therefore, cautious application and further research are needed.

## Figures and Tables

**Figure 1 cancers-15-04927-f001:**
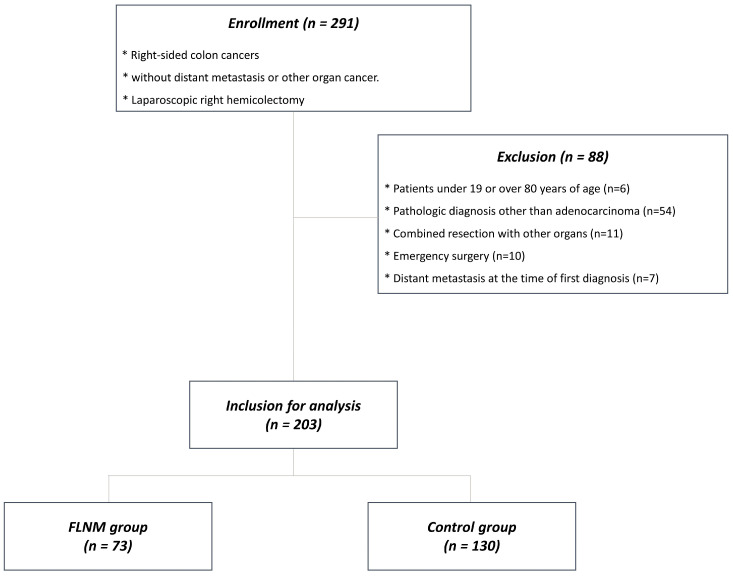
Flowchart of patient selection.

**Figure 2 cancers-15-04927-f002:**
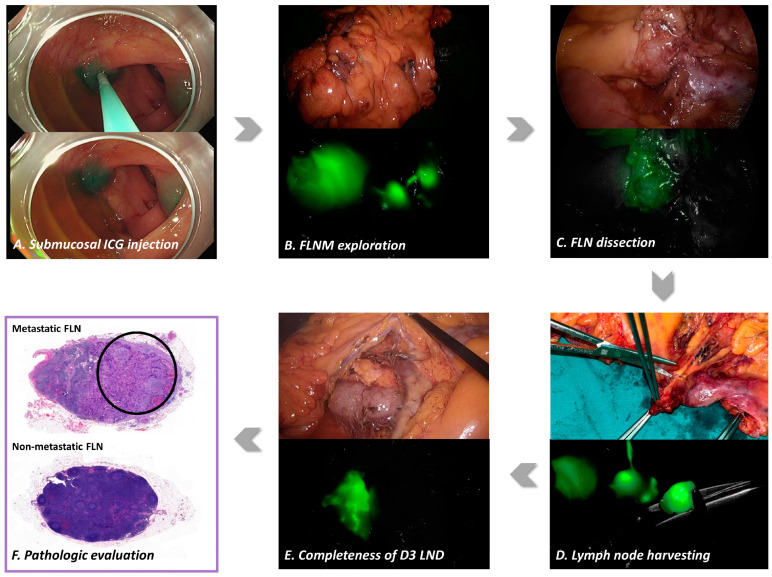
Standard protocol for fluorescence lymph node mapping (FLNM). (**A**) The endoscopic submucosal indocyanine green (ICG) injection procedure for tattooing was performed 12–18 h before surgery on two sites, as close as possible to the distal margin of the colon cancer. Indocyanine green injection protocols were standardized using an injection of 0.5–1 mL of minimal ICG concentration dose (0.25 mg/mL). (**B**) The FLNM exploration facilitated visualization of the tumor location, fluorescent lymphatic drainage pathway, and distribution of D3 lymph nodes under laparoscopic near-infrared (NIR) camera. (**C**) The suspicious metastatic lymph node was partially stained with ICG, which expressed weak fluorescent lymph node (FLN). (**D**) The D3 lymph nodes were harvested under the visualization provided by the laparoscopic NIR camera and labeled the fluorescence D3 lymph nodes, according to the colic arteries for pathologic assessment. (**E**) After the radical D3 lymph node dissection (LND) was completed, the absence of any residual FLN was confirmed to assess the completeness of the radical D3 LND. (**F**) Pathologic evaluations were conducted separately for the labeled fluorescent and non-fluorescent D3 lymph nodes to assess the presence of metastatic lymph nodes (hematoxylin and eosin staining, ×20). Black circle is the cancer infiltration area in the metastatic lymph node.

**Figure 3 cancers-15-04927-f003:**
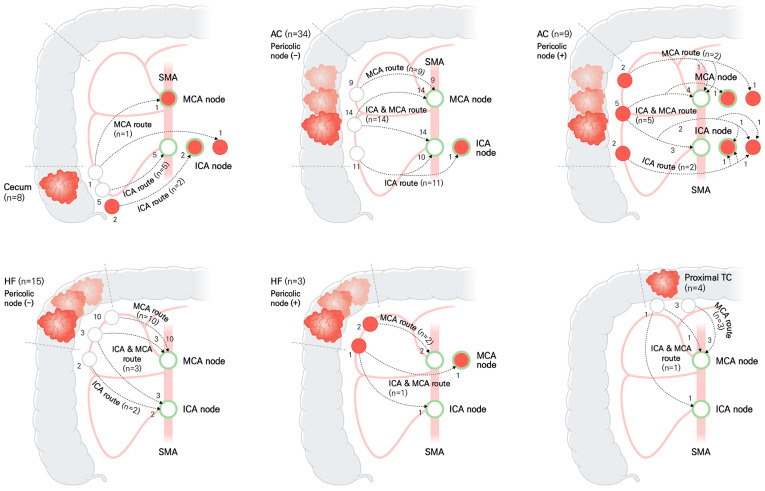
Schematic diagram illustrating the distribution of lymph node pathways and lymph node metastasis, according to the location of colon cancer in the FLNM group (*n* = 73). Green circle represents the fluorescent lymph node. Red ball is the metastatic lymph node. White ball is non-metastatic lymph node. The numbers next to the lymph nodes indicate the number of patients. Dotted line with arrow is the lymphatic drainage pathway. ICA, ileocolic artery; MCA, middle colic artery; SMA, superior mesenteric artery; AC, ascending colon; HF, hepatic flexure; TC, transverse colon.

**Figure 4 cancers-15-04927-f004:**
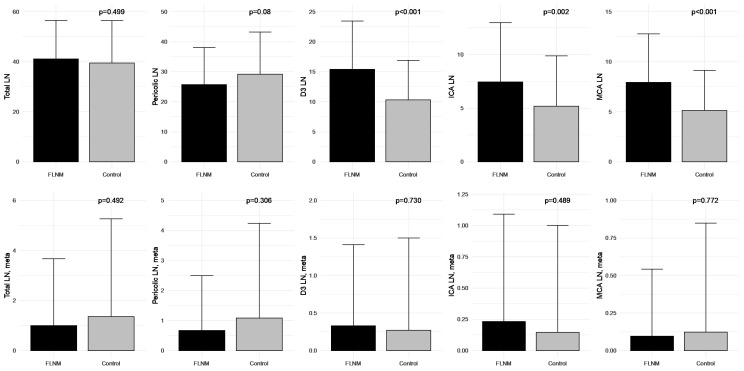
Comparison of the harvested and metastatic lymph node counts between fluorescence lymph node mapping (FLNM) (*n* = 73) and control (*n* = 130) groups. LN, lymph node; ICA, ileocolic artery; MCA, middle colic artery; meta, metastasis.

**Figure 5 cancers-15-04927-f005:**
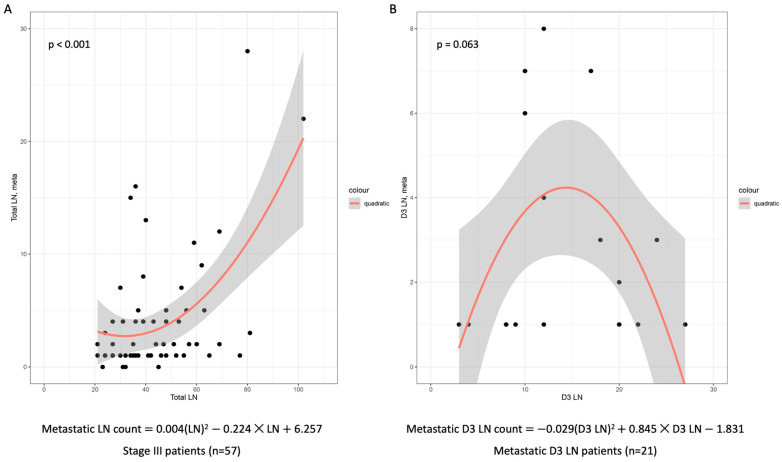
Scatter plot of the harvested and metastasis D3 lymph node counts in a regression analysis. (**A**) On the quadratic regression model, there was a statistically significant correlation between the harvested and metastatic lymph node counts in the stage III patients (*n* = 57). (**B**) The harvested D3 lymph node count showed a proportional correlation with the metastatic lymph node count of up to 15 lymph nodes. However, beyond 15 lymph nodes, the proportional correlation was reversed in patients with stage III disease having D3 lymph node metastasis (*n* = 21). Gray area represents 95% confidence interval of regression line. LN, lymph node.

**Figure 6 cancers-15-04927-f006:**
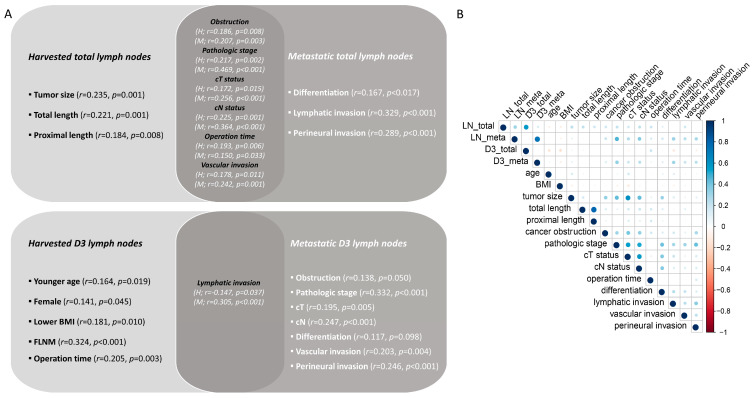
Clinicopathologic variables relating to harvested and metastatic lymph node counts analyzed using linear regression model (*n* = 203). (**A**) Correlation diagram for total and D3 lymph node counts. (**B**) Correlation plot for bivariate analysis representing the Spearman correlation coefficient between the lymph node counts and corresponding clinical factors. H, harvested lymph node count; M, metastatic lymph node count; BMI, body mass index; FLNM, fluorescence lymph node mapping; LN, lymph node.

**Table 1 cancers-15-04927-t001:** Patient’s characteristics (*n* = 203).

Clinical Variables		FLNM (*n* = 73)	Control (*n* = 130)
		*n* (%)	*n* (%)
Age, yr	mean ± SD	67.7 ± 11.8	67.6 ± 10.5
Sex	male	36 (49.3)	61 (46.9)
	female	37 (50.7)	69 (53.1)
BMI (kg/m^2^)	mean ± SD	23.5 ± 3.1	23.9 ± 2.8
Cancer location	cecum	8 (11.0)	19 (14.6)
	ascending colon	43 (58.9)	80 (61.5)
	hepatic flexure	18 (24.7)	19 (14.6)
	proximal transverse colon	4 (5.5)	12 (9.2)
cT status	cT 1–2	42 (57.5)	58 (44.6)
	cT 3–4	31 (42.5)	72 (55.4)
cN status	cN 0	48 (65.8)	51 (39.2)
	cN 1–2	25 (34.2)	79 (60.8)
pT status	pT 1–2	30 (41.1)	39 (30.0)
	pT 3–4	43 (58.9)	91 (70.0)
pN status	pN 0	57 (78.1)	89 (68.5)
	pN 1–2	16 (21.9)	41 (31.5)
Pathologic stage	I	29 (39.7)	34 (26.2)
	II	28 (38.4)	55 (42.3)
	III	16 (21.9)	41 (31.5)
Tumor size (cm)	mean ± SD	3.9 ± 2.2	4.4 ± 2.4
Differentiation	well	34 (46.6)	27 (20.8)
	moderate	37 (50.7)	95 (73.1)
	poorly	2 (2.7)	8 (6.2)
Cancer obstruction	positive	15 (20.5)	28 (21.5)
Lymphatic invasion	positive	13 (17.8)	26 (20.0)
Vascular invasion	positive	4 (5.5)	14 (10.8)
Perineural invasion	positive	14 (19.2)	26 (20.0)

FLNM, fluorescence lymph node mapping; SD, standard deviation; BMI, body mass index.

**Table 2 cancers-15-04927-t002:** Lymphatic drainage pattern and D3 lymph node metastasis in the FLNM group (*n* = 203).

		Cancer Location, *n* (%)				*p* Value
		Cecum	Ascending Colon	Hepatic Flexure	Proximal Transverse Colon	
FLNM group						
Lymphatic drainage pattern						
	ICA route	7 (87.5)	13 (30.2)	2 (11.1)	0	<0.001 ^†^
	MCA route	1 (12.5)	11 (25.6)	12 (66.7)	3 (75.0)	
	Dual route	0	19 (44.2)	4 (22.2)	1 (25.0)	
D3 LN metastasis						
	Negative	5 (62.5)	37 (86.0)	17 (94.4)	4 (100)	0.311 ^†^
	ICA LNs	2 (25.0)	4 (9.3)	0	0	
	MCA LNs	1 (12.5)	1 (2.3)	1 (5.6)	0	
	Both LNs	0	1 (2.3)	0	0	
Control group						
D3 LN metastasis						
	Negative	17 (89.5)	75 (93.8)	15 (78.9)	12 (100)	0.090 ^†^
	ICA LNs	1 (5.3)	3 (3.8)	0	0	
	MCA LNs	0	2 (2.5)	3 (15.8)	0	
	Both LNs	1 (5.3)	0	1 (5.3)	0	
Total						
D3 LN metastasis						
	Negative	22 (81.5)	112 (91.1)	32 (86.5)	16 (100)	0.125 ^†^
	ICA LNs	3 (11.1)	7 (5.7)	0	0	
	MCA LNs	1 (3.7)	3 (2.4)	4 (10.8)	0	
	Both LNs	1 (3.7)	1 (0.8)	1 (2.7)	0	

FLNM, fluorescence lymph node mapping; LN, lymph node; ICA, ileocolic artery; MCA, middle colic artery; ^†^: Fisher’s exact test.

**Table 3 cancers-15-04927-t003:** Lymph node (LN) metastasis in the stage III patients (*n* = 57).

Stage III		FLNM Group, *n* (%)	Control Group, *n* (%)	*p* Value
Percolic LNs metastasis	negative	2 (12.5)	0	0.075
	positive	14 (87.5)	41 (100)	
ICA LNs metastasis	negative	8 (50.0)	35 (85.4)	0.005
	positive	8 (50.0)	6 (14.6)	
MCA LNs metastasis	negative	12 (75.0)	34 (82.9)	0.496
	positive	4 (25.0)	7 (17.1)	
Total D3 LNs metastasis	negative	6 (37.5)	30 (73.2)	0.012
	positive	10 (62.5)	11 (26.8)	

FLNM, fluorescence lymph node mapping; LN, lymph node; ICA, ileocolic artery; MCA, middle colic artery.

**Table 4 cancers-15-04927-t004:** Lymphatic drainage pattern and D3 lymph node metastasis according to pericolic LN metastasis in the FLNM group (*n* = 73).

		D3 LN Metastasis				
		Negative	ICA LNs	MCA LNs	Both LNs	*p* Value
Pericolic LN metastasis; negative (*n* = 59)						
	ICA route	17 (29.8)	1 (100)	0	0	0.852 ^†^
	MCA route	22 (38.6)	0	0	1 (100)	
	Dual route	18 (31.6)	0	0	0	
Pericolic LN metastasis; positive (*n* = 14)						
	ICA route	0	4 (80.0)	0	0	0.039 ^†^
	MCA route	3 (50.0)	0	1 (50.0)	0	
	Dual route	3 (50.0)	1 (20.0)	1 (50.0)	1 (100)	
Total (*n* = 73)						
	ICA route	17 (27.0)	5 (83.3)	0	0	0.041 ^†^
	MCA route	25 (39.7)	0	1 (50.0)	1 (50.0)	
	Dual route	21 (33.3)	1 (16.7)	1 (50.0)	1 (50.0)	

FLNM, fluorescence lymph node mapping; LN, lymph node; ICA, ileocolic artery; MCA, middle colic artery, ^†^: Fisher’s exact test.

**Table 5 cancers-15-04927-t005:** D3 lymph node (LN) metastasis in the stage III patients of FLNM group (*n* = 16).

Stage III of FLNM Group		Fluorescent D3 LNs Metastasis, *n* (%)				
		Negative	ICA LNs	MCA LNs	Both LNs	Total
Non-fluorescent D3 LNs metastasis	Negative	6 (37.5) *	4 (25.0)	1 (6.3)	0	11 (68.8)
	ICA LNs	1 (6.3)	1 (6.3)	1 (6.3)	1 (6.3)	4 (25.0)
	MCA LNs	1 (6.3)	0	0	0	1 (6.3)
	Total	8 (50.0)	5 (31.3)	2 (12.5)	1 (6.3)	16 (100)

FLNM, fluorescence lymph node mapping; LN, lymph node; ICA, ileocolic artery; MCA, middle colic artery; * pericolic LNs metastasis without D3 LNs involvement.

**Table 6 cancers-15-04927-t006:** Univariate and multivariate analysis using simple and multivariable linear regression model for clinicopathologic factors relating to D3 lymph node metastasis (*n* = 203).

	Univariate			Multivariate		
**Harvested D3 LNs**						
**Variable**	**Beta**	**95% CI**	***p* Value**	**Beta**	**95% CI**	***p* Value**
age	−0.113	−0.205, −0.017	0.020	−0.112	−0.199, −0.026	0.011
sex (male)	−2.129	−4.161, −0.019	0.048	−2.645	−4.590, −0.699	0.008
BMI	−0.465	−0.818, −0.113	0.010	−0.451	−0.775, −0.126	0.007
operation time (min)	0.042	0.014, 0.069	0.003	0.031	0.003, 0.060	0.030
lymphatic invasion	−2.802	−5.431, −0.173	0.038	−2.359	−4.774, 0.057	0.056
FLNM	5.072	3.005, 7.138	<0.001	3.942	1.806, 6.078	<0.001
**Metastatic D3 LNs**						
**Variable**	**Beta**	**95% CI**	***p* Value**	**Beta**	**95% CI**	***p* Value**
cancer obstruction	0.398	0.003, 0.794	0.049			
pathologic stage	0.507	0.306, 0.707	<0.001	0.321	0.082, 0.559	0.009
cT status	0.212	0.064, 0.359	0.005			
cN status	0.365	0.163, 0.566	<0.001	0.200	−0.031, 0.432	0.089
differentiation	0.255	−0048, 0.558	0.099			
lymphatic invasion	0.910	0.515, 1.305	<0.001	0.682	0.284, 1.080	0.001
vascular invasion	0.839	0.277, 1.402	0.012			
perineural invasion	0.728	0.330, 0.324	<0.001			
FLNM	0.06	−0.281, 0.040	0.730	0.237	−0.088, 0.561	0.152

LN, lymph node; FLNM, fluorescence lymph node mapping; BMI, body mass index; CI, confidence interval.

**Table 7 cancers-15-04927-t007:** Univariate and multivariate analysis using simple and multivariable linear regression model for clinicopathologic factors relating D3 lymph node metastasis of stage III patients (*n* = 57).

	Univariate			Multivariate		
**Harvested D3 LNs**						
**Variable**	**Beta**	**95% CI**	***p* Value**	**Beta**	**95% CI**	***p* Value**
age	−0.108	−0.281, 0.065	0.216	−0.121	−0.210, −0.032	0.008
FLNM	7.567	3.706, 11.428	<0.001	−5.243	−7.278, −3.208	<0.001
length (total)	0.048	−0.083, 0.180	0.465	0.122	0.021, 0.222	0.018
length (proximal)	−0.013	−0.163, 0.136	0.858	−0.103	−0.218, 0.011	0.076
**Metastatic D3 LNs**						
**Variable**	**Beta**	**95% CI**	***p* Value**	**Beta**	**95% CI**	***p* Value**
sex (male)	−0.28	−1.377, 0.818	0.612	−0.969	−2.076, 0.138	0.085
operation time (min)	0.009	−0.006, 0.024	0.224	0.013	−0.002, 0.028	0.085
lymphatic invasion	1.486	0.407, 2.566	0.008	1.664	0.576, 2.752	0.003

LN, lymph node; FLNM, fluorescence lymph node mapping; CI, confidence interval.

## Data Availability

The data presented in this study are available on request from the corresponding author. The data are not publicly available due to ethical and privacy reasons.

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
