# Peer review of "Clinical Effectiveness of Fluorescence Lymph Node Mapping Using ICG for Laparoscopic Right Hemicolectomy: A Prospective Case–Control Study"

_cancers, 2023, doi:10.3390/cancers15204927_

Round 1

Reviewer 1 Report

The author investigated the clinical effectiveness of fluorescence lymph node mapping using ICG for laparoscopic right hemicolectomy from a prospective case-control study. The study draw the conclusion that FLNM can be effective in improving the oncological quality and completeness of D3 lymph node dissection in patients with stage III colon cancer during laparoscopic right hemicolectomy. Overall, the workload of this study is significant. I have some concerns as follows:

1.     Although there were higher number of harvested lymph nodes in the D3 area in FLNM group, but the proportion of metastatic lymph nodes among the harvested lymph nodes was similar in the FLNM and control groups, therefore, the value of FLNM still need to be verified. 

2.     In addition, lymph nodes filled full with tumor cells can not be detected by FLNM, so FLNM also had limitations in detecting metastatic lymph nodes. FLNM combined with experienced surgeon's visual inspection maybe a better manner for D3 lymph node dissection. 

3.     But for stage III patients, FLNM group had higher proportion of D3 lymph node metastasis compared to control group. The number of enrolled patients is relatively small in stage III group, which needs to expand the number of this subgroup to verify the conclusion. 

4.     Table 6 showed univariate and multivariate analysis for clinical factors predicting D3 lymph node metastasis. The author could also analyze the impact of clinical factors on the prediction of D3 lymph node metastasis in stage III group.    

Minor editing of English language required

Reviewer 2 Report

This is an observational study on the merits of ICG to improve the lymph node yield in CME right hemicolectomy for cancer. I have some comments.

1. The introduction is fine and concisely written. I do miss some discussion about the fact that some studies point to the fact that higher lymph node yield might also be a marker for immunogenic tumours, which could be a good prognostic feature.

2. The procedures and operations involved are well described. However, there's no mention about the indication for FLNM and selection of patients. Was this a surgeon preference? Was it something picked up over time by all surgeons involved? Were there differences in experience between surgeons performing FLNM and not? This needs some elaboration, especially the rate of FLNM over time.

3. The statistical analysis section needs to be elaborated. This is not a randomised trial, so there should consideration for confounding. This does not mean  just putting every parameter into a regression analysis,  but careful consideration in an e.g. causal diagram; for example, it doesn't make sense to include operation time in the regression if you are asking yourself a causal question, as FLNM increases operation time (thus time is a mediator).

4. How do you handle missing  data if there's any?

5. I would strongly discourage using p values in a Table 1 as this doesn't convey meaningful. information (depends on sample size; there are no attached hypotheses etc)

6. I am slightly surprised by the finding that the FLNM group had more ICA mets. When doing a proper CME, it is my impression that the ICA region is the easiest to access and clear, which wouldn't leave any room for improvement with FLNM. I might not understand the rationale here, but please comment. This is also slightly at odds with the baseline differences, where cN+ was much more common in the control group?

7. While the discussion is interesting, it is a very long read. There's also numerous insertions of results (mostly already shown) which should be omitted. I'd advise to shorten the discussion by half at least, trying not reproduce results and abbreviate the speculation.

8. The conclusions are quite definitely formulated. They should be softened, as there are numerous limitations to this study and it would be difficult to ascribe causality (FLNM cannot be said at this point to really "increase D3 node count by 50%", but could rather be framed as a promising strategy to potentially increase the nodal harvest, whilst the clinical benefit of this is still unclear).

Round 2

Reviewer 2 Report

Thank you for providing a revision to this paper. In general, I think this is an excellent paper with meticulous dissections and painstaking analysis on lymph node harvest. The responses are overall sound, but I have some concerns about my point 3 on statistical methods, where there seems too be a misunderstanding in all likelihood. I put forward an example where adjustment should not be made when answering a causal question (not adjusting for operation time) where the authors have done the complete opposite, it seems. Or does "operation time" refer to "year of operation"? This would indeed be a good thing to adjust for.

Moreover, the inserted phrase "The elimination criterion of covariance was set at <0.1) does not make sense to me. Perhaps the authors are using p values in a selection process? This is not advised when posing causal questions. I would rather see a causal diagram involved.

I would like to see a regression analysis with FLNM as exposure and lymph node harvest and/or lymph node metastasis as outcome, but with adjustment for numerous preoperative/baseline variables, such as: sex, bmi, cancer location, cT, cN, obstruction, operation year – these are potential true confounders, not covariates known after the fact (such as mediators - operation time - or pathological variables, unknown at the time of surgery). Table 6 and 7 might be interesting in some sense, but not from a causal point of view.

English is of a high quality.

Round 3

Reviewer 2 Report

I think it is time to stop tormenting the authors of this beautiful study. Although I would personally very much like to see a regression analysis as proposed by myself in previous round (comment 4), I wouldn't make this a prerequisite for publication. It would be entirely up to the editorial team.